# Modeling of Compressive Strength of Sustainable Self-Compacting Concrete Incorporating Treated Palm Oil Fuel Ash Using Artificial Neural Network

**Tawfiq Al-Mughanam [1], Theyazn H. H. Aldhyani [2] , Belal Alsubari [3,4] and Mohammed Al-Yaari [5,*]**

[1] Mechanical Engineering Department, King Faisal University, P.O. Box 380, Al-Ahsa 31982, Saudi Arabia; talmughanam@kfu.edu.sa

[2] Community College of Abqaiq, King Faisal University, P.O. Box 400, Al-Ahsa 31982, Saudi Arabia; taldhyani@kfu.edu.sa

[3] Department of Civil Engineering, Faculty of Engineering, University of Malaya, Kuala Lumpur 50603, Malaysia; alsubari@um.edu.my

[4] Department of Civil Engineering, Miami College of Henan University, Jinming Ave No.1, Kaifeng 475000, Henan, China

[5] Chemical Engineering Department, King Faisal University, P.O. Box 380, Al-Ahsa 31982, Saudi Arabia

[*] Correspondence: malyaari@kfu.edu.sa; Tel.: +966-13-589-8583

**Abstract:** The utilization of a high-volume of treated palm oil fuel ash (T-POFA) as a partial cement substitution is one of the solutions presented to reduce carbon dioxide emissions ($CO_2$) and improve concrete sustainability. In this study, the Adaptive Neuro-Fuzzy Inference System (ANFIS) is adapted as an artificial neural network (ANN) modeling tool to predict the compressive strength of self-compacting concrete (SCC) containing T-POFA. The ANFIS model has been developed and validated using concrete mixtures incorporating 0%, 10 wt%, 20 wt%, 30 wt%, 50 wt%, 60 wt%, and wt 70% T-POFA as a replacement of ordinary Portland cement (OPC) at a constant water/binder (W/B) ratio of 0.35. The experimental data were divided into 70% training data and 30% testing data. The experimental results of self-compacting concrete (SCC) containing T-POFA ensured comparable or higher compressive strengths, especially at later ages, when compared to the control SCC. However, the prediction results of the compressive strength of SCC samples using the ANFIS model are very close to the experimental values. The developed ANFIS model showed a highly-efficient performance to predict the SCC compressive strength. In addition, the obtained accurate predicted results using the developed ANN model will significantly affect the current experimental protocols, especially for costly and unsafe experiments.

**Keywords:** sustainability; treated palm oil fuel ash; self-compacting concrete; compressive strength; ANFIS model

## 1. Introduction

Our environment suffers from different serious issues, including the sustained increase in the emitted greenhouse gases and the generated solid wastes. During the calcination process of cement production, calcium carbonate ($CaCO_3$) is decomposed to form calcium oxide (CaO) and carbon dioxide ($CO_2$). In addition, the combustion of fossil fuels used in the cement production process produces huge quantities of carbon emissions [1]. Hence, cement production plants are considered one of the major producers of $CO_2$, with 5% of the global carbon emissions [2]. Concrete consumption is steadily increasing, and the annual production rate is approximately 1 ton per capita [3], and 1 ton of $CO_2$ is vented to the atmosphere per 1 ton of cement produced [4].

On the other hand, millions of tons of industrial and agricultural wastes are produced, but most of them are not recycled and thus cause many environmental sustainability problems. Oil palm trees (OPT) are cultivated in huge quantities in Asia, West Africa, and America, and the global generated OPT biomass was estimated to reach 110 Mt (dry basis) by 2020 [5].

Therefore, the potential use of rich cementitious wastes as a cement-substitute in concrete has attracted the researchers' attention. Supplementary cementitious materials (SCMs) have been introduced in the concrete mixtures for economic and environmental needs [6]. The sustained use of SCMs is a reflection of the construction bloom with economic and ethical-driven purposes. These additives are either natural or wastes of some industrial processes. In addition, different pozzolanic materials have been used. The use of different SCMs, including palm oil fuel ash [7], rice husk ash [8], fly ash [9], pulverized fuel ash [10], and date palm ash [11,12] were evaluated. It is confirmed that the use of those materials can improve the mechanical properties of concrete, including compressive strength [13–16].

Due to its distinguished rheological properties, self-compacting concrete has been used since 1988 [6] in Japan. As the name implies, it can be compacted under its weight with almost no vibration [17]. It offers some valuable benefits including, but not limited to, a quiet working environment (noise pollution reduction), and considerable reduction in cost (labor and time). Currently, it is widely used globally for different applications.

Recently, a new type of self-compacting concrete (SCC) with a high percentage of cement replacement by different SCMs has been improved. Although a high percentage (up to 80%) of cement replacement by fly ash is investigated [18,19], the use of palm oil fuel ash (POFA) with a lower percentage is studied [20–27]. It has been reported that POFA can be used as a cementitious material with an improvement of the concrete compressive strength in the long-term [28–32].

Pozzolanic materials such as POFA, produced in thermal power plants, are used in concrete to enhance strength and durability [33]. During the hydration process of Portland cement, calcium silica hydrate (C-S-H) and calcium hydroxide are produced. Then, with the availability of POFA, as a pozzolan, it reacts with calcium hydroxide to produce more C-S-H structures [34,35] and thus improve the overall quality of concrete [11]. These reactions can be typically expressed as follows:

$$2\ Ca_3SiO_5 + 7\ H_2O \rightarrow 3CaO.2SiO_2.4H_2O + 3\ Ca(OH)_2 + Heat \tag{1}$$

$$2\ SiO_2 + 3\ Ca(OH)_2 \rightarrow 3CaO.2SiO_2.3H_2O \tag{2}$$

Engineering problems have been frequently solved by artificial intelligence (AI) algorithms. Artificial neural networks (ANN) models, in particular, are among the most famous and powerful AI techniques [36]. Usually, engineering parameters are obtained experimentally, but with economical and/or safety concerns. Therefore, the application of AI (in general) and ANN (in particular) is promising and worth investigating.

One of the most used mechanical characteristics in concrete design is compressive strength. Therefore, extensive experimental work has been conducted to obtain this parameter for different types of concrete. However, due to the costly nature of such experimentations and a lack of generalized accurate empirical models, ANN models have been developed for different conditions and compositions. The compressive strength of conventional concrete [37–40], high-strength concrete [41–43], and concrete with silica fume [44,45], volcanic scoria [46], fly ash [45,47,48], palm oil fuel ash [21,49], clay bricks [50], limestone filler [51], and waste quartz mineral dust [52] have been predicted using ANN. Safiuddin et al. (2016) [49] developed an ANN model to predict the compressive strength of the SCC containing up to 30 wt% of POFA. Almost all available results have revealed that the developed ANN models are highly efficient for predicting the concrete compressive strength.

As discussed earlier, the development of a mathematical model to predict the compressive strength of a variety of concrete types and composition is targeted by many researchers. However, there is a lack of research on the use of ANN to model the compressive strength of SCC containing treated POFA, especially in high-volume substitutions. Therefore, this study aims to develop a highly efficient ANN model to predict the compressive strength of SCC containing low and high volumes of treated POFA as a cement replacement material.

## 2. Materials and Methods

### 2.1. Constituent Materials

Ordinary Portland cement (OPC) and treated palm oil fuel ash (T-POFA) are used in this study as binders. T-POFA has been prepared as described in previous work [53]. X-ray Fluorescence (XRF) was used to measure the chemical composition and loss on ignition (LOI) of OPC and T-POFA. The chemical compositions and some physical characteristics are presented in Table 1.

**Table 1.** Chemical compositions (wt%) and some physical characteristics of ordinary Portland cement (OPC) and treated palm oil fuel ash (T-POFA).

| Oxide Composition | OPC | T-POFA |
|---|---|---|
| $SiO_2$ | 17.60 | 69.02 |
| $Al_2O_3$ | 4.02 | 3.9 |
| $Fe_2O_3$ | 4.47 | 4.33 |
| CaO | 67.43 | 5.01 |
| MgO | 1.33 | 5.18 |
| $Na_2O$ | 0.03 | 0.18 |
| $K_2O$ | 0.39 | 6.9 |
| $SO_3$ | 4.18 | 0.41 |
| Others | 0.55 | 5.07 |
| $SiO_2 + Al_2O_3 + Fe_2O_3$ | 26.09% | 77.25% |
| Loss on ignition (LOI) | 2.4 | 1.8 |
| Specific surface area, BET ($m^2$/g) | 3.05 | 7.4 |
| Median particle size, $d_{50}$ (μm) | 21 | 13 |

The 4.75 mm maximum size local mining sand with water absorption of 1.13% and crushed stone with a maximum size of 12.5 mm and water absorption of 0.43% were used as fine aggregate (F.A.) and coarse aggregates (C.A.), respectively. An aqueous solution of modified polycarboxylate copolymers (Sika ViscoCrete-1600) was used as a superplasticizer to enhance workability and obtain the required fresh properties of the SCC following the EFNARC standard. It has a density of 1.09±0.02 kg/$m^3$, and the recommended dosage is between 0.5–2.0% by weight of cement.

### 2.2. Mix Proportions of Concretes

A total of seven different SCC mixes were prepared with 0%, 10%, 20%, 30%, 50%, 60%, and 70% (by weight) T-POFA substitutions of the ordinary Portland cement. Table 2 displays all SCC mix proportions. The content of the binder (B), water to binder (W/B) ratio, and superplasticizer (S.P.) are 480 kg/$m^3$, 0.35, and 1.3%, respectively. The SCCs mixes were prepared and intensely discussed elsewhere [53,54].

**Table 2.** Mix proportions for self-compacting concretes (SCCs) containing T-POFA.

| Mix Code | Cement (kg/m³) | Water (kg/m³) | W/B Ratio | T-POFA | | Fine Aggregate (F.A.) (kg/m³) | Coarse Aggregate (C.A.) (kg/m³) | Superplasticizer (%) |
|---|---|---|---|---|---|---|---|---|
| | | | | (kg/m³) | (wt%) | | | |
| SCC0 | 480 | 168 | 0.35 | 0 | 0 | 925 | 760 | 1.3 |
| SCC10 | 432 | 168 | 0.35 | 48 | 10 | 923 | 760 | 1.3 |
| SCC20 | 384 | 168 | 0.35 | 96 | 20 | 948 | 770 | 1.3 |
| SCC30 | 336 | 168 | 0.35 | 144 | 30 | 944 | 770 | 1.3 |
| SCC50 | 240 | 168 | 0.35 | 240 | 50 | 925 | 758.2 | 1.3 |
| SCC60 | 192 | 168 | 0.35 | 288 | 60 | 925 | 758.2 | 1.3 |
| SCC70 | 144 | 168 | 0.35 | 336 | 70 | 925 | 758.2 | 1.3 |

*2.3. Preparation and Testing of Concretes*

The fresh properties of SCC such as slump flow, $T_{50}$ cm, V-funnel, J-ring, L-box segregation resistance were examined according to EFNARC (2002). All the fresh properties were in the range specified by the European code for SCC (EFNARC, 2002) [55]. The compressive strength experiments were carried out at 1, 3, 7, 28, 56, and 90 days, following BS EN 12390-3:2002 (2002) [56].

*2.4. Adaptive Neuro-Fuzzy Inference System (ANFIS) Model*

The ANFIS model is an ANN algorithm proposed by Jang [57]. It is used to solve complicated and nonlinear problems. It is a powerful hybrid algorithm, which consists of a neural network and fuzzy logic. The ANFIS model can be used for data prediction by using an adaptive system in the input layer to obtain the optimal membership function. In this work, the ANFIS model has been used to predict the compressive strength due to the small size of the dataset. This model is more appropriate for small datasets than other ANN models. Furthermore, it has the benefit of having numerical and linguistic knowledge.

The ANFIS model has five significant layers, namely the fuzzification layer, the antecedent layer, the strength normalization layer, the consequent layer, and the inference layer [58]. These layers have numerous nodes, known by the node function. The structure of the ANFIS model with two input parameters and an output parameter is schematically represented in Figure 1. The if-then rules have been employed as follows:

$$Rule1 : if \ x \ is \ A_1 \ and \ y \ is \ B_1, \ then \ f_1 = p_1 x + q_1 y + r_1 \tag{3}$$

$$Rule1 : if \ x \ is \ A_2 \ and \ y \ is \ B_2, \ then \ f_1 = p_2 x + q_2 y + r_2 \tag{4}$$

where $A_1$, $A_2$, $B_1$, and $B_2$ are fuzzy set, and $x$, and $y$ are the input parameters for node $i$. The $p_1$, $p_2$, $q_1$, $q_2$, $r_1$, and $r_2$ are the consequent parameters. The output of the ANFIS model is the parameter $f$.

**Layer 1 (Fuzzification layer):**

The first layer uses the membership function to convert the inputs into a fuzzy set. In the present study, the Gaussian membership function is employed due to its simplicity, with a low number of parameters as compared with the generalized bell-shape membership function.

$$O_{1,i} = \mu \ A_i(x) \ for \ i = \ 1, \ 2 \tag{5}$$

$$O_{1,i} = \mu \ B_i(y) \ for \ i = \ 1, \ 2 \tag{6}$$

$$\mu \ A_i(x_1) = \frac{1}{1 + \left(\frac{x - c_i}{\sigma_i}\right)^{2b_i}} \tag{7}$$

where:

$\mu(x)$ and $\mu(y)$ are membership functions;

$A_i$ represent the linguistic variable; and

$\sigma_i$, $b_i$, and $c_i$ are the parameters of the Bell function.

Figure 1 shows 7 simple inputs like SCC0, SCC10, SCC20, SCC30, SCC50, SCC60, and SCC70. These inputs are converted to fuzzy values by using the membership function. The fuzzy rules are generated using membership functions.

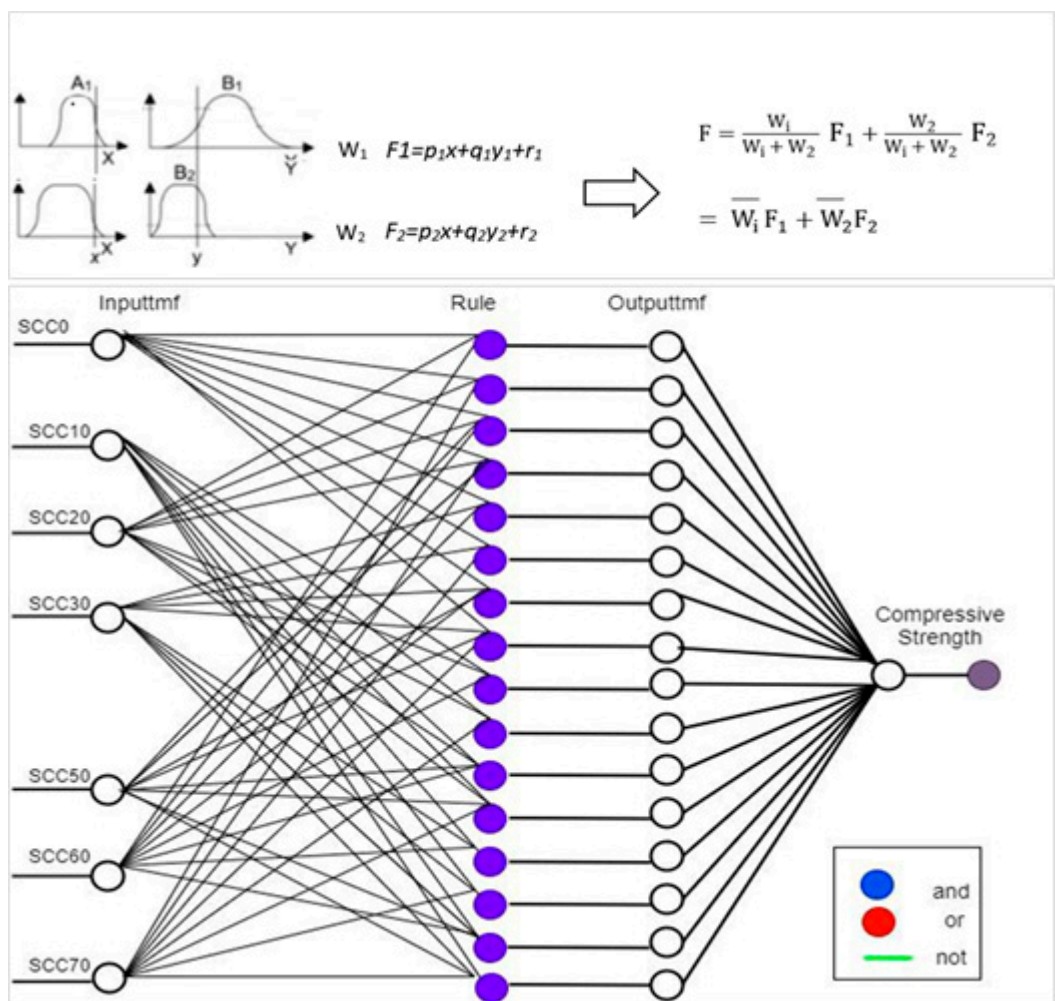

**Figure 1.** Architecture of the Adaptive Neuro-Fuzzy Inference System (ANFIS) model.

**Layer 2 (Antecedent layer):**

The node of the second layer is a fixed node. The node value is multiplied by the input signals to serve as an output signal.

$$O_{2,i} = w_i = \mu A_i(x) * \mu B_i(y), \; i = 1, 2 \tag{8}$$

where $w_i$ signal represents the firing strength of the rule.

**Layer 3 (Strength Normalization Layer):**

Firing strength is normalized by computing the ratio of the $i_{th}$.

$$O_{3,i} = \overline{w}_i = \frac{w_i}{w_{1+w_2}}, \; i = 1, 2 \tag{9}$$

where the $O_{3,i}$ is the output of Layer 3 and $\overline{w}$ is the normalized firing-strength.

**Layer 4 (Consequent Layer):**

The node of the fourth layer is adaptable, and the output of this layer is $O_{4,i}$. The node function of this layer is explained as follows:

$$O_{4,i} = \overline{w}_i.f_i = \overline{w}_i.(p_i x + q_i y + r_i) \tag{10}$$

where $p_i$, $q_i$, and $r_i$ are consequent parameters used for the fuzzy inference system function ($f_i$).

**Layer 5 (Inference Layer):**

This layer is used to obtain the overall output. It can be described as:

$$O_{5,i} overall\ output = \sum \overline{w}_i f_i = \frac{\sum_i \overline{w}_i f_i}{\sum_i w_i} \tag{11}$$

The ANFIS is a back-propagation algorithm, by which the output signals from layer 5 are computed backward to the input nodes in layer 1.

In the present study, the ANFIS model is proposed to predict the compressive strength of concrete incorporating T-POFA. The proposed model has been developed using MATLAB. Seven mix variables, namely the cement content (kg/m$^3$), water content (kg/m$^3$), W/B ratio, T-POFA content, F.A. content (kg/m$^3$), C.A. content (kg/m$^3$), and S.P. (% B) have been considered as input parameters for the proposed model to predict the compressive strength. The dataset has been divided into 70% for training and 30% for testing. The ANFIS model has two approaches: Scatter partition, and grid partition. In this work, scatter partition has been employed due to the small size of the dataset. The Scatter partition approach uses the clustering approach to split the dimension vectors in the specific area of the fuzzy rules. In this study, more focus on the use of the Fuzzy C-Means (FCM) clustering and back-propagation algorithms was given.

*2.5. Performance Measurement*

Several statistical analyses, like mean square error (MSE), root mean square error (RMSE), mean error, and Pearson's correlation coefficient (R), have been used to evaluate the performance of the developed model. The used statistical parameters were defined as follows:

- Mean Square Error (MSE):

$$MSE = \frac{1}{N} \sum_{i=1}^{N} (y - \hat{y})^2 \tag{12}$$

- Root Mean Square Error (RMSE):

$$RMSE = \sqrt{\sum_{i=1}^{N} \frac{(y - \hat{y})^2}{N}} \tag{13}$$

- Mean Error:

$$MeanError = (y - \hat{y}) \tag{14}$$

where:

- ○　$y$, and $\hat{y}$ are the predicted and the experimental responses, respectively; and
- ○　N is the total number of variables.

- % Regression Correlation Coefficient (R%):

$$R\% = \frac{n\left(\sum xy\right) - \left(\sum x\right)\left(\sum y\right)}{\left[n\sum(x^2) - \sum(x^2)\right]\left[n\sum(y^2) - \sum(y^2)\right]} \times 100\% \tag{15}$$

where:

- ○　　R: Pearson's correlation coefficient;
- ○　　x: Input values of the first set of training data;
- ○　　y: Input values of the second set of training data; and
- ○　　n: Total of simple input data.

## 3. Results and Discussion

### 3.1. Compressive Strength of Concretes

The experimental values of the compressive strength of the tested SCCs incorporating T-POFA are shown in Table 3. First, SCCs contained T-POFA are divided into low-volume (10%, 20%, and 30%) and high-volume (50%, 60%, and 70%) replacement levels. At the early-ages of 1, 3, and 7 days, all the SCCs contained low-volume T-POFA have exhibited higher or comparable compressive strength in comparison to the control specimen. This could be attributed to the filler effects of the T-POFA. The higher fineness of the T-POFA has acted as a filler between the gaps and voids of the concrete, resulting in higher compressive strength [28].

**Table 3.** Compressive strength of the tested SCCs incorporating T-POFA.

| Sample No. | 1 Day (MPa) | 3 Days (MPa) | 7 Days (MPa) | 28 Days (MPa) | 56 Days (MPa) | 90 Days (MPa) |
|---|---|---|---|---|---|---|
| | 39.0 | 51.2 | 57.5 | 67.1 | 70.5 | 72.0 |
| **SCC0** | 40.5 | 50.8 | 58.5 | 68.1 | 71 | 74 |
| | 37.5 | 51.6 | 56.5 | 66.1 | 69 | 70 |
| | 38.2 | 56.0 | 63.2 | 69.0 | 77.0 | 80.6 |
| **SCC10** | 40 | 55 | 64 | 71 | 76 | 81.2 |
| | 36.6 | 57 | 62.4 | 67 | 78 | 79.8 |
| | 36.0 | 54.6 | 61.8 | 73.0 | 86.0 | 88.0 |
| **SCC20** | 37.5 | 55.2 | 62.8 | 74 | 87 | 87.5 |
| | 34.5 | 53.8 | 60.8 | 71 | 85 | 88.5 |
| | 33.5 | 47.0 | 63.2 | 71.7 | 84.5 | 86.2 |
| **SCC30** | 34 | 49 | 64 | 70.7 | 85 | 87 |
| | 33 | 45 | 62.4 | 72.7 | 84 | 85.4 |
| | 28.0 | 40.0 | 52.0 | 69.0 | 75.2 | 78.4 |
| **SCC50** | 29 | 42 | 53 | 67 | 76 | 79 |
| | 27 | 38 | 51 | 71 | 74.4 | 77.8 |
| | 17.0 | 29.0 | 50.0 | 68.0 | 74.7 | 76.6 |
| **SCC60** | 18 | 31 | 52 | 68.5 | 75.7 | 77.6 |
| | 16 | 27 | 48 | 67.5 | 73.7 | 75.6 |
| | 14.0 | 26.0 | 47.0 | 65.5 | 72.9 | 74.5 |
| **SCC70** | 14.5 | 27.5 | 49 | 66.5 | 73.9 | 75.5 |
| | 13.5 | 24.5 | 45 | 64.5 | 71.9 | 73.5 |

However, all the SCC mixes incorporated high-volume T-POFA in early-ages of the curing have shown compressive strength reductions in comparison to the SCC without T-POFA (control sample). The main reason for the strength reductions at the early-ages is due to the massive dilution of cement content, which has caused less hydration products, namely, calcium hydrate silicate (C–H–S) [59]. The compressive strength reductions of SCC are overcome by increasing the curing time. As the curing of SCCs continues, the reaction between calcium hydroxide $Ca(OH)_2$ and the silica from T-POFA have

taken place to form additional C–H–S products, resulting in a better bonding between the paste and aggregates which leads to enhance the compressive strength of SCCs.

### 3.2. Prediction of the Compressive Strength by the ANFIS Model

#### 3.2.1. Training of the ANFIS Model

Seventy percent of the dataset has been considered for the training process. The results of the proposed model for the training state were very superior for the prediction of the compressive strength. Table 4 shows the predicted compressive strength results obtained by the ANFIS model during the training phase. It is observed that the prediction results are the same as the experimental results (R% = 100%). This implies the highly efficient performance of the developed model. Figure 2 illustrates the histogram error of the predicted values at the training state. The error histogram metrics are used to find the errors between the predicted values and the target values. As these error values specify how the predicted values deviate from the target values, these values can be negative. The maximum RMSE was reported to be $5.4899 \times 10^{-05}$.

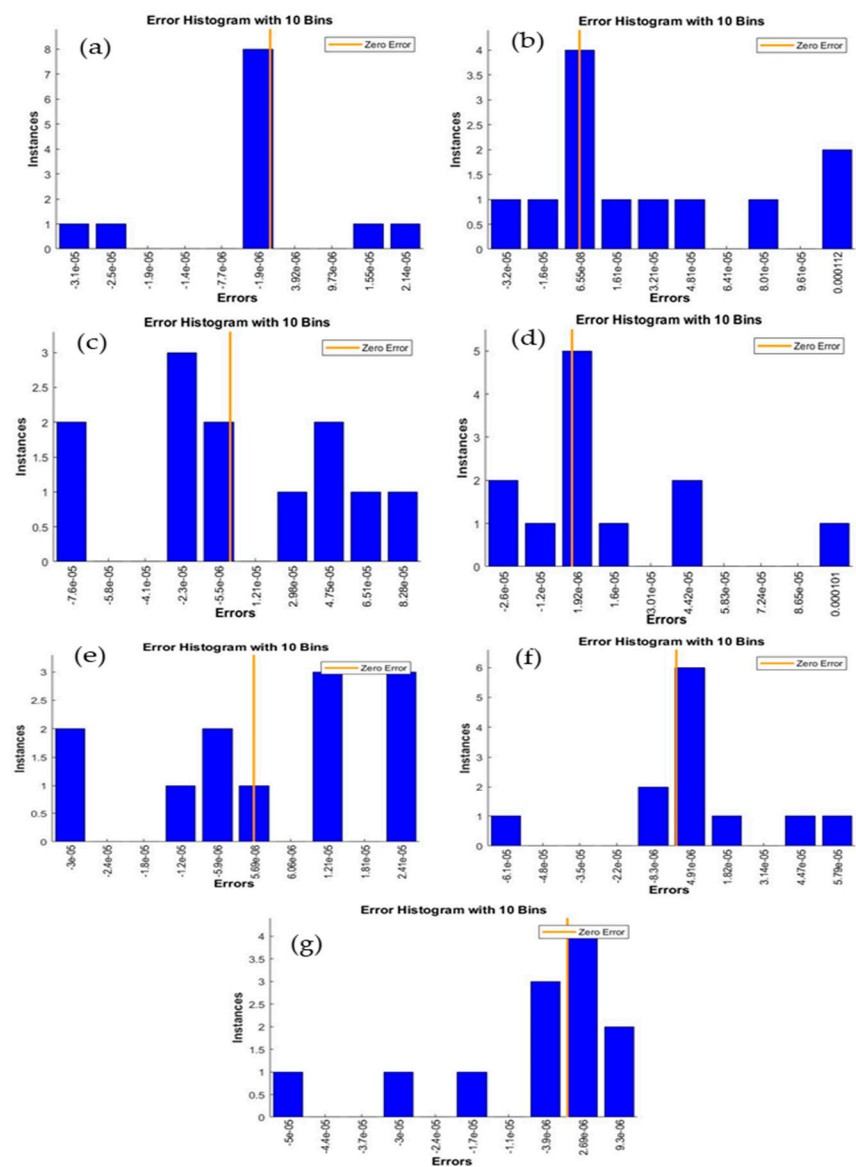

**Figure 2.** Histogram error of the ANFIS model for the training data of (**a**) SCC0, (**b**) SCC10, (**c**) SCC20, (**d**) SCC30, (**e**) SCC50, (**f**) SCC60, and (**g**) SCC70.

**Table 4.** Performances of the ANFIS model in the training phase.

| Dataset | Training Dataset | | | |
| --- | --- | --- | --- | --- |
| | MSE | RMSE | Mean Error | R (%) |
| SCC0 | $4.4450 \times 10^{-10}$ | $1.083 \times 10^{-05}$ | $1.0581 \times 10^{-06}$ | 100 |
| SCC10 | $1.5337 \times 10^{-09}$ | $3.9162 \times 10^{-05}$ | $3.16 \times 10^{-06}$ | 100 |
| SCC20 | $3.0139 \times 10^{-09}$ | $5.4899 \times 10^{-05}$ | $5.115 \times 10^{-06}$ | 100 |
| SCC30 | $1.5174 \times 10^{-09}$ | $3.8954 \times 10^{-05}$ | $1.2936 \times 10^{-05}$ | 100 |
| SCC50 | $3.6809 \times 10^{-10}$ | $1.9186 \times 10^{-05}$ | $1.6667 \times 10^{-06}$ | 100 |
| SCC60 | $9.557 \times 10^{-10}$ | $3.0914 \times 10^{-05}$ | $5.04 \times 10^{-06}$ | 100 |
| SCC70 | $3.7251 \times 10^{-10}$ | $1.9301 \times 10^{-05}$ | $7.3697 \times 10^{-06}$ | 100 |

### 3.2.2. Testing of the ANFIS Model

The testing process is used to validate the ANFIS developed model to predict the compressive strength of SCCs at 90 days by using another dataset. Table 5 presents the predicted values at the testing state. According to the evaluation metrics (MSE, RMSE, mean error, and R), the predicted values of the compressive strength are very close to the experimental ones.

**Table 5.** Performances of the ANFIS model in the testing phase.

| Dataset | Testing Dataset | | | |
| --- | --- | --- | --- | --- |
| | MSE | RMSE | Mean Error | R (%) |
| SCC0 | 0.4490 | 0.6701 | 0.0951 | 98.54 |
| SCC10 | 0.2439 | 0.4938 | 0.2658 | 88.85 |
| SCC20 | 0.1701 | 0.4124 | 0.3283 | 87.04 |
| SCC30 | 0.7396 | 0.8600 | 0.1443 | 94.04 |
| SCC50 | 0.7657 | 0.8751 | 0.0449 | 98.48 |
| SCC60 | 0.2035 | 0.4511 | 0.3896 | 98.72 |
| SCC70 | 0.8540 | 0.9241 | 0.9022 | 99.34 |

In addition, Figure 3 shows the regression plot for the predicted values of the compressive strength during the training and testing states. This plot is used to find the relationship between the predicted and the actual values by using Pearson's correlation. While the target (x-axis) values represent the experimental data, the output (y-axis) values represent the predicted values obtained by the ANFIS developed model. Results presented in Figure 3 revealed the highly efficient performance of the developed model.

A comparison between the experimental and predicted values of the compressive strength at the curing time of 90 days is presented in Table 6. The percentages of the mean errors were very low (in the range between 0.022–1.06%), which indicate the high robustness of the developed model to predict the compressive strength.

**Table 6.** Comparison between the experimental and predicted values of compressive strength at 90 days.

| Curing Time (Days) | Sample Code | Compressive Strength (MPa) | | |
| --- | --- | --- | --- | --- |
| | | Experimental Values | Predicted Values | Mean Errors |
| | SCC0 | 72 | 72.14 | 0.14 |
| | SCC10 | 80 | 81.39 | 0.79 |
| | SCC20 | 88.0 | 88.02 | 0.022 |
| Day 90 | SCC30 | 86.2 | 87.033 | 0.83 |
| | SCC50 | 78.4 | 78.02 | 0.37 |
| | SCC60 | 76.6 | 75.65 | 1.61 |
| | SCC70 | 74.5 | 75.56 | 1.06 |

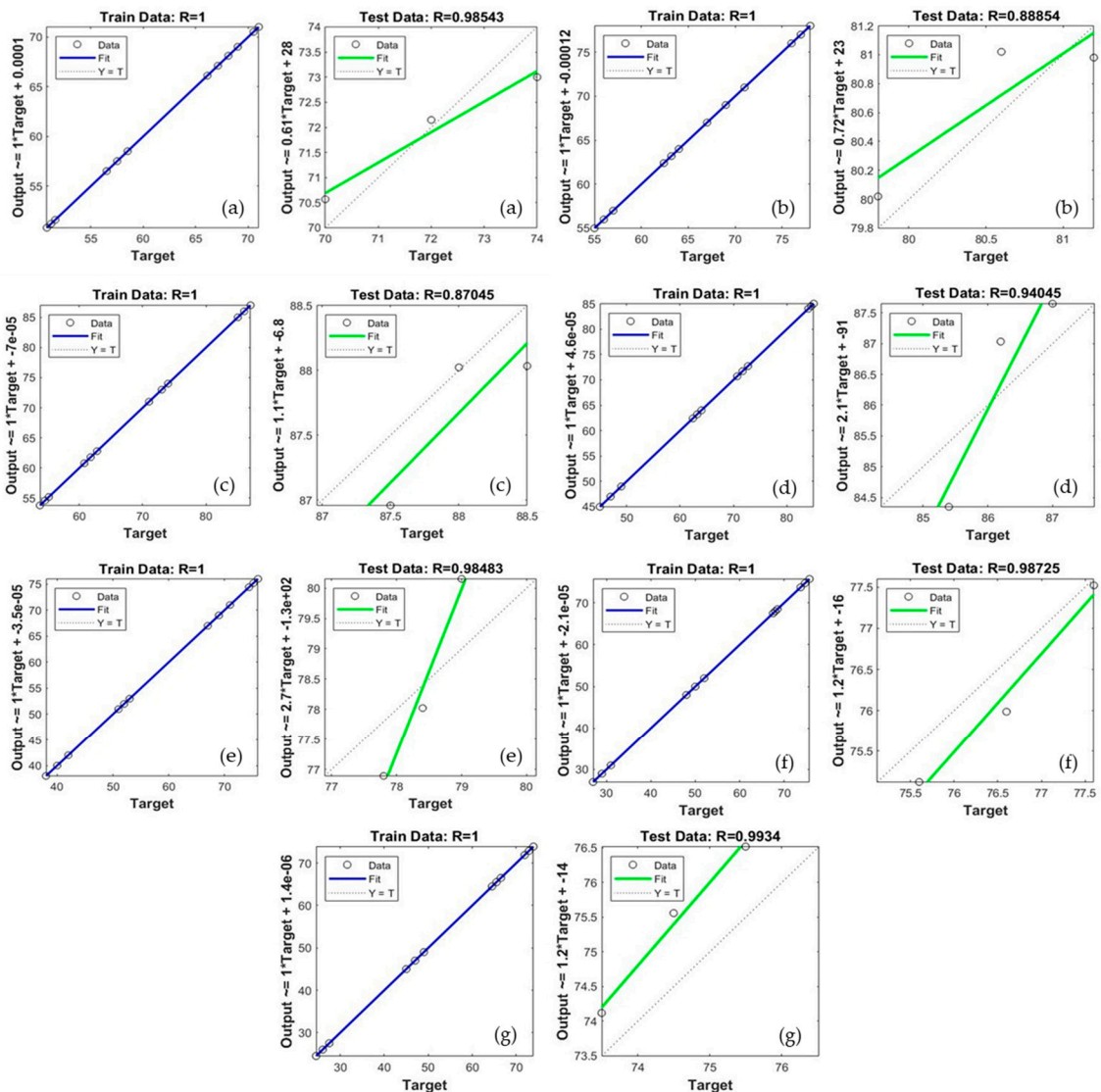

**Figure 3.** Regression plot for the predicted values of the compressive strength of SCCs with different T-POFA content (wt%): (**a**) 0, (**b**) 10, (**c**) 20, (**d**) 30, (**e**) 50, (**f**) 60, and (**g**) 70.

## 4. Conclusions

The following conclusions can be drawn for the SCC containing T-POFA and the developed ANFIS model used to predict the concrete compressive strength.

a.  SCC containing low-volume T-POFA has exhibited comparable or higher compressive strengths in early-ages and later-ages when compared to SCC control.

b.  Compressive strengths of SCC incorporating high-volume T-POFA were lower than the reference SCC sample, however, with increased curing time, the compressive strengths were similar or higher than the SCC without T-POFA (control sample).

c.  The predicted results of the developed ANFIS model used to predict the compressive strengths of SCC were very close to the experimental values of SCC.

d.  Based on the predicted results of compressive strength, it is approved that the developed ANFIS model has successfully modeled the compressive strengths of SCC at different conditions.

**Author Contributions:** All authors contributed significantly to the completion of this article, but they had different roles in all aspects. Conceptualization, T.A.-M., T.H.H.A., B.A. and M.A.-Y.; Data curation, T.H.H.A. and B.A.; Formal analysis, T.A.-M., T.H.H.A., B.A. and M.A.-Y.; Funding acquisition, T.A.-M.; Investigation, T.H.H.A. and B.A.; Methodology, T.A.-M., T.H.H.A., B.A. and M.A.-Y.; Project administration, M.A.-Y.; Software, T.H.H.A.; Validation, B.A.; Visualization, M.A.-Y.; Writing—original draft, T.H.H.A. and B.A.; Writing—review & editing, M.A.-Y. All authors have read and agreed to the published version of the manuscript.

**Funding:** This research and the APC were funded by the Deputyship for Research & Innovation, Ministry of Education in Saudi Arabia through the project number IFT20146.

**Acknowledgments:** The authors extend their appreciation to the Deputyship for Research & Innovation, Ministry of Education in Saudi Arabia for funding this research work through the project number IFT20146.

**Conflicts of Interest:** The authors declare no conflict of interest.

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
