# Peer review of "Modeling of Compressive Strength of Sustainable Self-Compacting Concrete Incorporating Treated Palm Oil Fuel Ash Using Artificial Neural Network"

_sustainability, doi:10.3390/su12229322_

Round 1

Reviewer 1 Report

The article is well structured and very readable.

The attached PDF file contains questions and suggestions, which I hope will be used to improve the publication.

Unfortunately, despite a very good impression, the publication contains deficiencies that prevent a full assessment of the content. The following questions need to be answered:
- how many samples were applied for training of each variant (SCC10, SCC20, ....SCC70),
-from how many mixtures the samples came from,
-how many batches of materials were used during the tests.

Without this information, the results obtained cannot be judged. Some justifications related to the chosen model are also required (for more details, see PDF).

Author Response

Dear Respected Reviewer,

Thanks with best regards

Reviewer 2 Report

The manuscript focus on the application of the Adaptive Neuro-Fuzzy Inference System as an artificial neural network modeling tool to predict the compressive strength of self-compacting concrete containing high-volume of treated palm oil fuel ash. I think that there is potential in this paper. However, revision is necessary to increase the quality of the article:

- Lines 53-54. I agree that the use of different SCA, including silica fume, ground granulated blast-furnace slag, fly ash, date palm ash, and rice husk ash was evaluated in [6-11]. However, the use of citation pockets like [6-11] does not add much to the literature survey. Each reference in the article should be cited separately with the description what is done in cited paper and why this paper has been cited,

- lines 82-84. I agree that compressive strength of conventional concrete, high-strength concrete and concrete with silica fume, volcanic scoria, fly ash, palm oil fuel ash, clay bricks and limestone filler has been predicted using ANN. However, in this place one important attempt of the prediction of the compressive strength of concrete modified with high volume of waste quartz mineral dust has been missed (https://www.sciencedirect.com/science/article/pii/S0959652618337612). I suggest to add this paper to the literature survey,

- please describe in the article how the characteristics presented in Table 1 were determined. Were they taken from the XRD or declared by the manufacturer?

- Table 1 – please add the row “others” to show the other elements (currently the sum of the elements does not give 100%),

- please provide more data about used superplasticizer.

Author Response

(The authors gave the same response as above.)

Reviewer 3 Report

Theme of manuscript Modeling of Compressive Strength of Sustainable Self-Compacting Concrete Incorporating Treated Palm Oil Fuel Ash Using Artificial Neural Network is very interesting and topical; some minor concerns need to be addressed before it can be considered for publication.

  • have You any idea to use any other methods different modeling tool than ANN match to Your research ?
  • there is lack of citations (line 72-81)
  • could You explain why did You prepare mixtures with presented amount of T-POFA ? (line 104)

Generally, this work is very-well written and I do not any more negative comments. 

Author Response

(The authors gave the same response as above.)

Reviewer 4 Report

This paper used ANFIS modelling tool to predict the compressive strength of SCC containing different contents of T-POFA (0%, 10%, 20%, 30%, 50%, 60% and 70%). The proposed model has been developed using MATLAB. Seven mix variables, namely the cement content, water content, W/B ratio, T-POFA content, F.A. content, C.A. content, and S.P. have been considered as input parameters for the proposed model to predict the compressive strength. The dataset has been divided into 70% for training and 30% for testing. How many dataset have been used for this training? The small size of the dataset was not sufficient for the prediction of the compressive strength. The robustness of the modelling should be a great issue. In the current version, it is not suitable to be published in the journal.

Here are some specific comments.

  1. Line 49, Supplementary cementitious additives (SCA), generally in the concrete industry called supplementary cementitious materials (SCMs).
  2. Line 59, “If” should be changed to “It”.
  3. Table 2, S.P. (% should add “(%)”.
  4. Line 62, “SSC” should be “SCC”.
  5. Line 123 error in blue.
  6. Line 107, please add the SP quantity.
  7. What is the method used to the treatment of POFA? Please give more details. What about the rheological proprieties of concrete mixes? Since the high fineness of the T-POFA, without the additional quantity of SP, the rheological proprieties of concrete mixes could be influenced by the high water demand of T-POFA.

Author Response

(The authors gave the same response as above.)

Round 2

Reviewer 4 Report

After reading the revised manuscript, the issues have been modified and explained adequately. Therefore, I would like to recommend it to publish in the journal.